# Assessing the Long-Term Creep Behaviour of Hydrothermally Treated Japanese Cedar Wood Using the Short-Term Accelerated Stepped Isostress Method

**DOI:** 10.3390/polym15204149

**Published:** 2023-10-19

**Authors:** Jin-Wei Xu, Cheng-Chun Li, Jian-Wei Liu, Wen-Chao Chang, Wen-Shao Chang, Jyh-Horng Wu

**Affiliations:** 1Department of Forestry, National Chung Hsing University, Taichung 402, Taiwan; ecsgunro@gmail.com (J.-W.X.); alaninfire@gmail.com (C.-C.L.); ted2301636@gmail.com (J.-W.L.); 2Tainan District Agricultural Research and Extension Station, Council of Agriculture, Tainan 712, Taiwan; wcchang@mail.tndais.gov.tw; 3Lincoln School of Architecture and the Built Environment, University of Lincoln, Lincoln LN6 7TS, UK; WChang@lincoln.ac.uk; 4Advanced Plant and Food Crop Biotechnology Center, National Chung Hsing University, Taichung 402, Taiwan

**Keywords:** creep behaviour, hydrothermal treatment, Japanese cedar (*Cryptomeria japonica*), short-term accelerated test, stepped isostress method

## Abstract

In this study, short-term accelerated creep tests were conducted using the stepped isostress method (SSM) to investigate the impact of hydrothermal treatment on the long-term creep behaviour of Japanese cedar wood and to determine optimal hydrothermal treatment conditions. The results showed that SSM can effectively predict the creep behaviour of hydrothermally treated wood. Among the treatment conditions tested, Japanese cedar wood treated hydrothermally at 180 °C for 4 h exhibited higher flexural strength retention (91%) and moisture excluding efficiency (MEE) (44%) and demonstrated superior creep resistance compared to untreated wood. When subjected to a 30% average breaking load (ABL) over 20 years, the specimen’s creep compliance, instantaneous creep compliance, *b* value, activation volume, and improvement in creep resistance (ICR) were 0.17 GPa^−1^, 0.139 GPa^−1^, 0.15, 1.619 nm^3^, and 4%, respectively. The results indicate that subjecting Japanese cedar wood to hydrothermal treatment at 180 °C for 4 h has a negligible effect on its flexural properties but results in significant improvements in both dimensional stability and creep resistance.

## 1. Introduction

Wood continues to be an important material for humans; it is used in construction, decorating, furniture-making, and various other applications due to its excellent strength-to-weight ratio, workability, attractive appearance, and ability to regulate environmental humidity and temperature [1]. However, wood is susceptible to moisture-related damage from various organisms and microorganisms [2]. Therefore, numerous studies in the past have focused on physical and chemical modifications that will extend the lifespan of wood. Among these methods, heat treatment has been widely adopted worldwide because it requires no chemical agents, is more environmentally friendly than other methods, and is a convenient process [3]. Heat treatment is primarily performed at temperatures ranging from 160 to 240 °C and involves the degradation of hemicellulose and hydroxyl groups to enhance the dimensional stability and biological resistance of wood. The process varies depending on factors such as treatment temperature, heating rate, holding time, heating medium, pressure, and wood species. Different heat treatment conditions and reaction media can also affect changes in various chemical functional groups. For example, high-temperature heat treatment with hot air can cause hydroxyl groups to esterify into carbonyl groups and the reaction affects the physical and mechanical properties of wood [4]. Hydrothermal treatment is considered a relatively mild heat treatment method. It involves high temperatures (160–240 °C) and high pressures (1–3.5 MPa) with water as the reaction medium and the self-ionized hydronium ions (H_3_O^+^) formed in water catalyse the reactions [5]. Hydrothermal treatment involves only water as a reaction medium and does not require the addition of chemical agents. This not only reduces equipment corrosion but also lowers production costs and expenses, making it an environmentally friendly wood treatment technology.

Current research on heat-treated wood primarily focuses on common physical and mechanical properties. However, there is a lack of research on the creep behaviour of heat-treated wood, which is important because heat-treated wood used in construction is often subjected to long-term loads and may creep over time. Many researchers have attempted to use accelerated tests to simulate the results of long-term creep through short-term creep test models and predict the long-term creep behaviour of materials. Consequently, many researchers have developed accelerated creep testing methods, such as the time–temperature superposition principle (TTSP), the stepped isothermal method (SIM), the time–stress superposition principle (TSSP), and the stepped isostress method (SSM) [6]. Among these methods, SSM is less affected by temperature and humidity gradients within a test material. In our previous research, we successfully used SSM to analyse wood-inorganic composites [7]. Additionally, in Japan, Japanese cedar (*Cryptomeria japonica*) wood is commonly used in long-term creep conditions for beams, columns, bridges, and ships [8]. Consequently, there is a considerable amount of literature discussing the creep behaviour of Japanese cedar wood [9,10] or the impact of different treatments on the creep behaviour of Japanese cedar wood [3,11]. However, there is currently no literature on the application of SSM on hydrothermally treated Japanese cedar wood. Therefore, in this study, we aimed to investigate the long-term creep behaviour of hydrothermally treated Japanese cedar wood under different treatment conditions and to find the optimal hydrothermal treatment conditions. Furthermore, the results of SSM and 90-day experimental creep tests were used to investigate the suitability of SSM to assess the long-term creep behaviour of hydrothermally treated Japanese cedar wood. Overall, the results of this study are conducive to predicting the long-term creep behaviour of untreated and hydrothermally treated wood in a short period and can be used to optimise the modification of treatment conditions.

## 2. Materials and Methods

### 2.1. Materials

Japanese cedar (*Cryptomeria japonica* D. Don) (40–45 years old) timber, which is the most common domestically produced wood in Taiwan, was supplied by Jang Chang Lumber Industry Co., Ltd. (Hsinchu, Taiwan). Flat-sawn timbers with dimensions of 500 mm × 136 mm × 26 mm and an average density of 396 kg/m^3^ were prepared from conventionally kiln-dried wood with an average moisture content of 14%. Before hydrothermal treatment, all specimens were conditioned at 20 °C and 65% relative humidity (RH) for at least 2 weeks.

### 2.2. Hydrothermal Treatment of Japanese Cedar Timber

In this study, the specimens with dimensions of 500 mm × 136 mm × 26 mm, along with distilled water (approximately 4 L) equivalent to half their dry weight, were placed in a semi-industrial heat treatment chamber (San Neng Ltd., Chiayi, Taiwan) equipped with thermocouple and pressure sensors. The temperature was increased at a rate of 3 °C/min to reach 160, 180, and 200 °C, and the specimens were held at those temperatures for 4, 8, and 16 h, respectively. Finally, after the chamber had cooled to room temperature, the hydrothermally treated Japanese cedar wood specimens treated under different conditions were placed in a 20 °C and 65% RH environment for subsequent analysis.

### 2.3. Characterization of Wood Properties

To determine the physicomechanical properties of the untreated and hydrothermally treated Japanese cedar wood, several properties, including equilibrium moisture content (EMC), moisture excluding efficiency (MEE), and flexural properties, were determined. According to CNS 452 [12] and ASTM D1037 [13], the EMC and MEE of the specimens with cubic dimensions of 26 mm × 26 mm × 26 mm were measured using Equations (1) and (2), respectively. Flexural properties were determined according to CNS 454 [14] via a universal testing machine (Shimadzu AG-10kNX, Tokyo, Japan). In brief, the modulus of rupture (MOR) and modulus of elasticity (MOE) of the specimens with dimensions of 500 mm × 136 mm × 26 mm were determined via a three-point static bending test with a loading speed of 5 mm/min and a span of 364 mm. The MOR and MOE were calculated using Equations (3) and (4), and all tests were carried out in an air-conditioned room at 20 °C. Thirteen replicate specimens were used for each determination.
(1)$$\mathrm{EMC}\;\left(\%\right)=\frac{\left(W_{c}\;–{\;W}_{0}\right)}{{\;W}_{0}}\times \;100$$
where W_c_ represents the mass (g) of specimen conditioned at 20 °C/65% RH for two weeks and W_0_ represents the mass (g) of oven-dried specimen.
(2)$$\mathrm{MEE}\;\left(\%\right)=\frac{\left({\mathrm{EMC}}_{t}\;–{\;\mathrm{EMC}}_{0}\right)}{{\;\mathrm{EMC}}_{0}}\times \;100$$
where EMC_0_ and EMC_t_ represent the EMC (%) of untreated and hydrothermally treated Japanese cedar wood, respectively.
(3)$$\mathrm{MOR}\;\left(\mathrm{MPa}\right)=\frac{3\mathrm{PL}}{2bh^{2}}$$
(4)$$\mathrm{MO}E\left(\mathrm{GPa}\right)=\frac{\Delta PL^{3}}{4\Delta \mathrm{Yb}h^{3}}$$
where P is the maximum load (N), L is the length (mm) of the span, ΔP is the difference of load (N) between 10% and 40% values of the maximum load, ∆Y is deflection (mm) due to ∆P, b is the width (mm) of the specimen, and h is the thickness (mm) of the specimen.

### 2.4. Creep Behaviour Analysis

In this study, a 90-day creep test was conducted on hydrothermally treated specimens prepared under different conditions. During the test, specimens with dimensions of 230 mm (L) × 50 mm (W) × 10 mm (H) were placed on a creep testing rig, and the average breaking load (ABL) of untreated specimens at 30% was used as the creep load. The test was carried out with a centrally concentrated load at 20 °C and 65% RH. Over the 90-day test period, a data acquisition system was used to continuously monitor creep deflection at the centre of the span of the specimens. According to ASTM D2990 [15] standards, the creep strain of the specimens was calculated using Equation (5):(5)$$\mathrm{Creep}\;\mathrm{strain}\;\left(\%\right)=\frac{6h}{L^{2}}\;\times \;\Delta I\times \;100$$
where h is the thickness of the specimen (mm), L is the span length (mm), and ∆*I* is creep deflection (mm).

In addition, the SSM testing approach, based on Hadid et al. [16] and Giannopoulos and Burgoyne [17], was employed to conduct short-term accelerated creep tests on hydrothermally treated specimens. During the tests, specimens with dimensions of 230 mm (L) × 50 mm (W) × 10 mm (H) were placed in a universal strength testing machine (Shimadzu AG-10kNX, Tokyo, Japan). The tests were conducted under an ambient temperature of 20 °C with a centrally concentrated load. The initial load for the tests was set as a reference load at 30% of the average breaking load (ABL) of untreated specimens. Subsequently, six different stress intervals and test durations, including 5% ABL/2 h, 5% ABL/3 h, 5% ABL/5 h, 7.5% ABL/2 h, 10% ABL/2 h, and 12.5% ABL/2 h, were applied to investigate the effect of SSM on the creep master curve. Furthermore, the obtained SSM master curves under various conditions were verified with the results of the 90-day experimental creep tests. The creep deflection was transformed into creep compliance using Equation (6) [18] to gain a more in-depth understanding of the predicted master curve for hydrothermally treated Japanese cedar wood.
(6)$$S(t)\;({\mathrm{MPa}}^{-1})=\frac{4bh^{3}D(t)}{PL^{3}}$$
where *S*(*t*) is the time-dependent compliance (MPa^−1^), b is the specimen width (mm), h is the specimen thickness (mm), *D*(*t*) is the creep deflection at time *t* (mm), and P is the load (N). Subsequently, the master curves were modelled using the Findley power equation, Equation (7), as follows [19]:*S*(*t*) = *S*_0_ + *ae^bt^*(7)
where *S*_0_ is the instantaneous elastic compliance, *a* and *b* are constant numbers, and *t* is the elapsed time.

The improvement in creep resistance (ICR) was calculated using Equation (8) to estimate the creep resistance of the hydrothermally treated Japanese cedar wood under long-term conditions.
(8)$$\mathrm{ICR}\;\left(\%\right)=[1-\frac{{S\left(t\right)}_{h}}{{S\left(t\right)}_{u}}]\;\times \;100$$
where *S*(*t*)_u_ and *S*(*t*)_h_ are the time-dependent creep compliance values (MPa^−1^) of untreated and hydrothermally treated Japanese cedar wood, respectively. Moreover, the activation volume (*V**) was computed using the Eyring model presented in Equation (9). This model has been effectively employed to evaluate the creep behaviour of wood-based composites and estimate the shift factor (*α*_σ_), which demonstrates the rate of expression with stress level, as indicated by our previous studies [6,7].
(9)$$\log\;\alpha_{\sigma }=\frac{V*(\sigma -\sigma_{\mathrm{ref}})}{(2.303\;\times 10^{27}\;\times \;\mathrm{kT})}$$
where *V** is the activation volume (m^3^), *α*_σ_ is the horizontal shift factor, σ_ref_ and σ denote the reference stress and test stress (MPa), respectively, *k* is the Boltzmann constant (1.38 × 10^−23^ J/K), and *T* is the absolute temperature (K).

### 2.5. Analysis of Variance

The significance of differences was calculated by one-way ANOVA with Scheffe’s post hoc test, and *p* values < 0.05 were considered significant.

## 3. Results and Discussion

### 3.1. Impact of Different Hydrothermal Treatment Conditions on the Physicomechanical Properties of Japanese Cedar Wood

Many studies have indicated that heat treatment can effectively reduce moisture absorption in wood and improve its dimensional stability, thereby extending its service life. At the same time, heat treatment may influence the creep properties of wood due to variations in equilibrium moisture content or flexural properties [3]. Table 1 shows the impact of different hydrothermal treatment conditions on the physicomechanical properties of Japanese cedar wood. It can be observed that the untreated wood had an EMC of 13.9% at 20 °C/65% RH. However, the EMC of Japanese cedar wood subjected to hydrothermal treatment decreased with increasing temperature and time of treatment, ranging from 5.4% to 7.9%. The reduction in EMC became more significant in response to higher temperatures and longer treatment times, showing a significant difference from untreated wood. This indicates that hydrothermal treatment effectively reduces the EMC of Japanese cedar wood, possibly due to the thermal degradation of hemicellulose and lignin during hydrothermal treatment, leading to a reduction in hydroxyl groups in the wood. This trend supports the results of Kutnar and Kamke [20]. Furthermore, the wood subjected to hydrothermal treatment exhibited significantly improved moisture resistance, with resistance increasing with the increase in treatment temperatures and times, ranging from 44% to 62%. This demonstrates that hydrothermal treatment effectively enhances the dimensional stability of wood. This improvement is likely due to a reduction in the hydroxyl groups that occurred as a result of the thermal degradation of hemicellulose and lignin and the formation of microcracks from lignin thermal degradation, which affected the wood’s moisture resistance [21].

In addition, Table 1 shows that the MOE (modulus of elasticity) of untreated wood was 6.8 GPa. Among the hydrothermal treatment groups, only the 200 °C/16 h treatment exhibited a significant difference compared to untreated wood, with its MOE dropping to 4.9 GPa. For most of the hydrothermally treated wood, there was no significant difference in MOE compared to untreated wood, with values ranging from 5.8 to 6.8 GPa. This could be attributed to the lower treatment temperature employed in this study compared to previous research and the thermal degradation and rearrangement of amorphous cellulose, resulting in increased crystallinity which in turn affects the MOE [22]. In addition, as shown in Table 1, it can be observed that the MOR of untreated cedar was 59.7 MPa. The MOR of hydrothermally treated wood decreased with increasing temperature and duration of treatment, and the extent of reduction became more pronounced at higher treatment temperatures and longer treatment times, ranging from 29.1 to 58.7 MPa. Among these treatments, 160 °C/4 h, 160 °C/8 h, 160 °C/16 h, 180 °C/4 h, and 180 °C/8 h showed no significant difference compared to untreated wood, while 200 °C/16 h exhibited the lowest MOR. This result is similar to the results obtained in previous studies of heat-treated wood and can be primarily attributed to the thermal degradation of hemicellulose and cellulose, which leads to a reduction in plasticity and damage to the crystalline region of wood [23,24].

### 3.2. Impact of Different Hydrothermal Treatment Conditions on the Creep Properties of Japanese Cedar Wood

To investigate the impact of hydrothermal treatment conditions on the creep properties of Japanese cedar wood, in this study, short-term accelerated creep tests were conducted using SSM on untreated wood and hydrothermally treated wood with no significant difference in MOR compared to untreated wood. The hydrothermal treatments included 160 °C/4 h, 160 °C/8 h, 160 °C/16 h, 180 °C/4 h, and 180 °C/8 h. Additionally, the effectiveness of the SSM was verified using the results of a 90-day experimental creep test, as shown in Figure 1. It can be observed that the creep master curves were not significantly affected by differences in the SSM tests, and the trends in the SSM master curves were similar to those of the 90-day experimental creep curves. This indicates that SSM is suitable for assessing the long-term creep behaviour of hydrothermally treated Japanese cedar wood. Furthermore, the creep strain of samples treated at 160 °C for 4 (Figure 1B), 8 (Figure 1C), and 16 h (Figure 1D) was higher than that of the untreated wood (Figure 1A), indicating that cellulose molecules were more mobile under the 160 °C treatment conditions, thus resulting in a higher free volume. This trend is in agreement with the results of compression creep analysis reported by Kutnar and Kamke [20] on heat-treated hybrid poplar wood. This may be due to the thermal degradation of hydrogen bonds between hemicellulose and cellulose, causing relaxation of the crystalline cellulose chains and leading to a higher creep strain [24,25]. In addition, Figure 2 displays the SSM master curves and 90-day experimental creep curves for untreated (Figure 2A) and hydrothermally treated wood (Figure 2B–F). The SSM master curves followed a similar trend and had comparable creep rates to the experimental creep curves. This test result aligns with findings of Hadid et al. [16], supporting the effective application of the SSM in predicting and assessing the long-term creep behaviour of hydrothermally treated Japanese cedar wood.

Figure 3 presents the predicted long-term creep compliance curves for untreated and hydrothermally treated Japanese cedar wood using the SSM. It can be observed that the creep compliance of untreated wood and wood hydrothermally treated at 180 °C for 4 and 8 h was similar, and their trends were also similar. In contrast, the creep compliances of wood hydrothermally treated at 160 °C for 4, 8, and 16 h were higher than those of untreated wood. Among them, the 160 °C/4 h treatment showed the highest creep compliance, consistent with the trend in creep strain observed. In addition, from the predicted creep compliance using the SSM shown in Figure 3, it can be observed that the trends in each group conformed with the Findley power law equation. Therefore, in this study, the creep compliance was further fitted with the Findley power law equation, and the creep resistance under different hydrothermal treatment conditions was evaluated. Table 2 shows that the *b* value of untreated wood was 0.27, and the predicted creep compliance at 1, 5, 10, 20, and 50 years were 0.16, 0.16, 0.17, 0.17, and 0.18 GPa^−1^, respectively. In contrast, the *b* values for wood treated hydrothermally at 160 °C for 4, 8, and 16 h were higher than those of untreated wood, indicating that the 160 °C hydrothermal treatment condition was unfavourable for long-term creep properties. This result is consistent with the higher creep strain and free volume observed in the wood treated hydrothermally at 160 °C. Conversely, the *b* values for wood treated at 180 °C for 4 and 8 h were lower than those of untreated wood, suggesting better long-term creep properties. However, the instant elastic creep compliance of wood treated at 180 °C for 8 h was higher than that of untreated wood. Hence, as shown in Figure 3, when the predicted time exceeds 35 years, the creep compliance of wood treated at 180 °C for 8 h becomes lower than that of untreated wood. This may be attributed to the thermal degradation of hemicellulose and cellulose, which causes a reduction in wood plasticity and damage to the crystalline region. Among all the specimens, only the wood treated at 180 °C for 4 h exhibited both lower instantaneous elastic compliance and a lower *b* value. At predicted creep times of 1, 5, 10, 20, and 50 years, its creep compliance was 0.15, 0.16, 0.16, 0.17, and 0.17 GPa^−1^, respectively. The corresponding improvement rates in creep resistance were 3%, 3%, 4%, 4%, and 5%. This indicates that the hydrothermal treatment condition of 180 °C for 4 h was effective in enhancing creep resistance.

Furthermore, the activation volume was calculated in this study using the Eyring equation for untreated and hydrothermally treated wood. Figure 4 shows that all groups exhibited a high degree of linearity, with regression analysis *R*^2^ values exceeding 0.90, indicating that the horizontal displacements for each group conformed to the Eyring equation and that the creep mechanisms for all groups were similar. Furthermore, the activation volume calculated from the Eyring equation revealed that among all the groups, the wood treated at 180 °C for 4 h had the lowest activation volume (1.619 nm^3^). This result indicates that wood hydrothermally treated at 180 °C for 4 h exhibited the best creep resistance. In contrast, the activation volume for untreated wood was 1.636 nm^3^, while for wood treated hydrothermally at 160 °C/4 h, 160 °C/8 h, 160 °C/16 h, and 180 °C/8 h, the activation volumes were higher than that of untreated wood, measuring 2.051, 1.879, 2.107, and 1.904 nm^3^, respectively. This suggests that Japanese cedar wood undergoes hydrothermal alteration, leading to a relaxation of the molecular chains in the crystalline region, which results in an increase in activation volume [3,25].

## 4. Conclusions

In this study, Japanese cedar (*Cryptomeria japonica*), one of the most common plantation tree species in Taiwan, was utilised as the wood material, and hydrothermal treatments were applied at different temperatures (160, 180, and 200 °C) and durations (4, 8, and 16 h) using a semi-industrial reaction vessel. In addition to investigating the impact of hydrothermal treatment on the fundamental properties of Japanese cedar wood, the SSM was also applied to predict the long-term creep behaviour of hydrothermally treated wood. The results showed that the MEE of Japanese cedar wood increased with high hydrothermal treatment temperatures and long treatment times. Conversely, the EMC decreased with increasing hydrothermal treatment temperature and time. Regarding flexural properties, the MOE and MOR generally decreased with increasing hydrothermal treatment temperature and time. Notably, the MOE and MOR for specimens treated at 160 °C for 4, 8, and 16 h and 180 °C for 4 and 8 h did not significantly differ from those of untreated wood. In addition, the SSM results demonstrated a high degree of consistency in the master curves obtained under different test conditions. Additionally, these curves exhibited trends similar to the changes observed in the 90-day experimental creep test curves, indicating the effective applicability of the SSM in predicting the long-term creep behaviour of untreated and hydrothermally treated Japanese cedar wood. Among all the hydrothermal treatments, the specimens treated at 180 °C for 4 h exhibited the best creep resistance in Japanese cedar wood. This study provides a systematic, fast, and convenient evaluation method for understanding the creep properties of materials. At the same time, it also offers a concrete and practical reference for selecting optimal treatment conditions for wood processing or modification with the aim of improving the creep performance of wood.

## Figures and Tables

**Figure 1 polymers-15-04149-f001:**
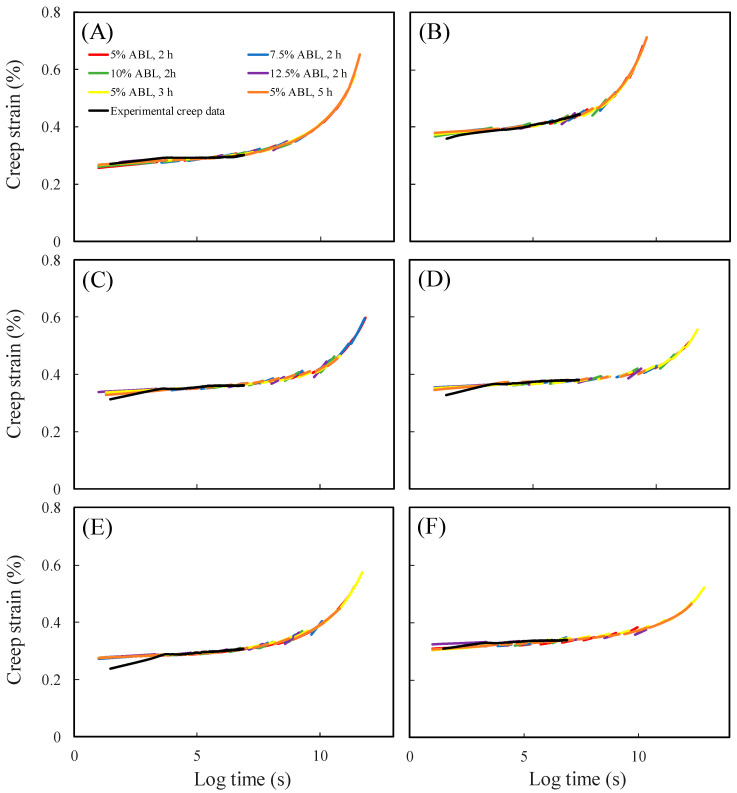
SSM master curves and 90-day experimental creep data of untreated (**A**), 160 °C/4 h (**B**), 160 °C/8 h (**C**), 160 °C/16 h (**D**), 180 °C/4 h (**E**), and 180 °C/8 h (**F**) hydrothermally treated Japanese cedar wood.

**Figure 2 polymers-15-04149-f002:**
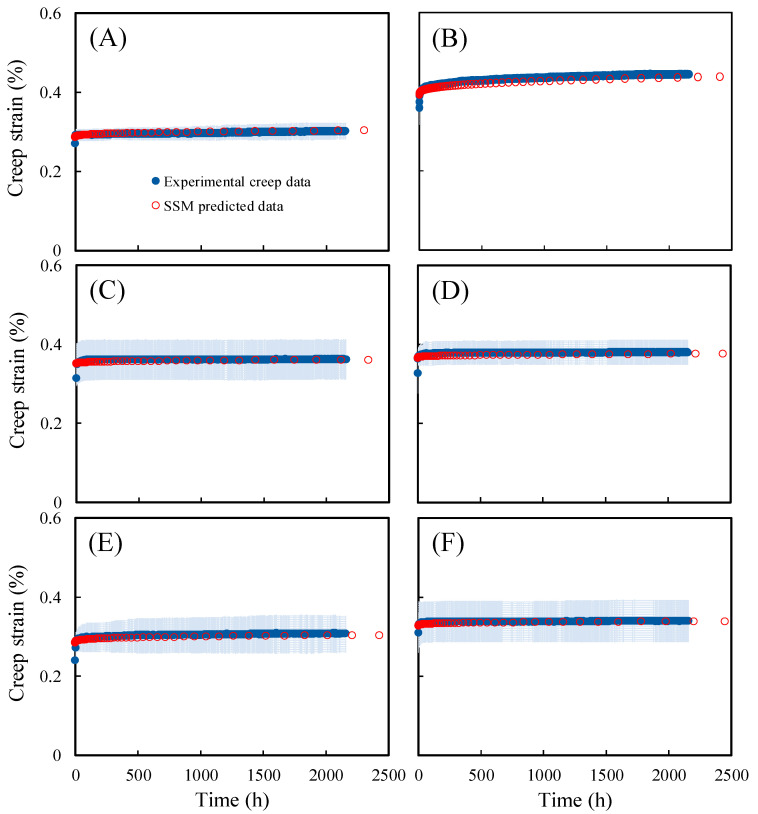
Comparison of SSM master curves and 90-day experimental creep data of untreated (**A**), 160 °C/4 h (**B**), 160 °C/8 h (**C**), 160 °C/16 h (**D**), 180 °C/4 h (**E**), and 180 °C/8 h (**F**) hydrothermally treated Japanese cedar wood. Experimental creep data are the mean ± SD (light blue ribbon) (*n* = 2).

**Figure 3 polymers-15-04149-f003:**
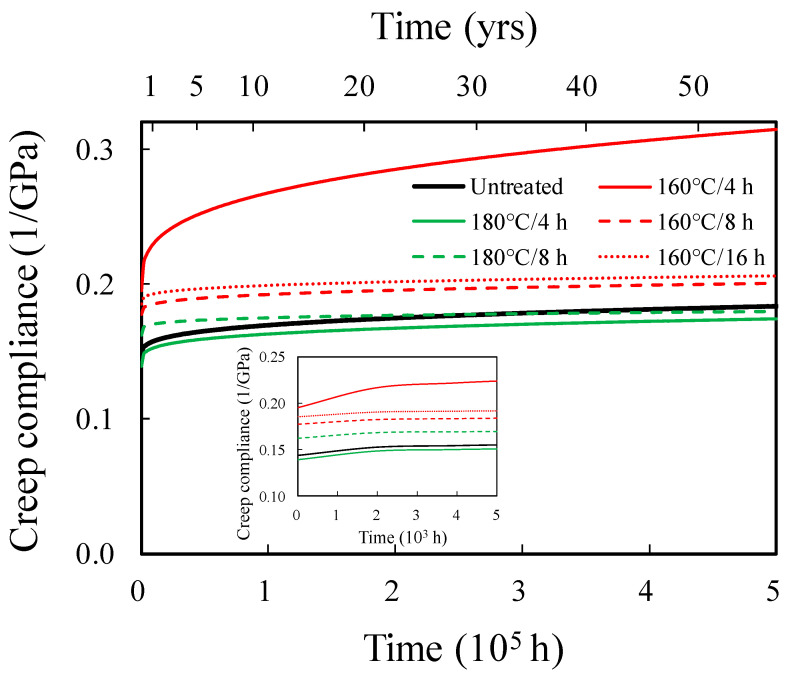
SSM predicted creep data of untreated, 160 °C/4 h, 160 °C/8 h, 160 °C/16 h, 180 °C/4 h, and 180 °C/8 h hydrothermally treated Japanese cedar wood.

**Figure 4 polymers-15-04149-f004:**
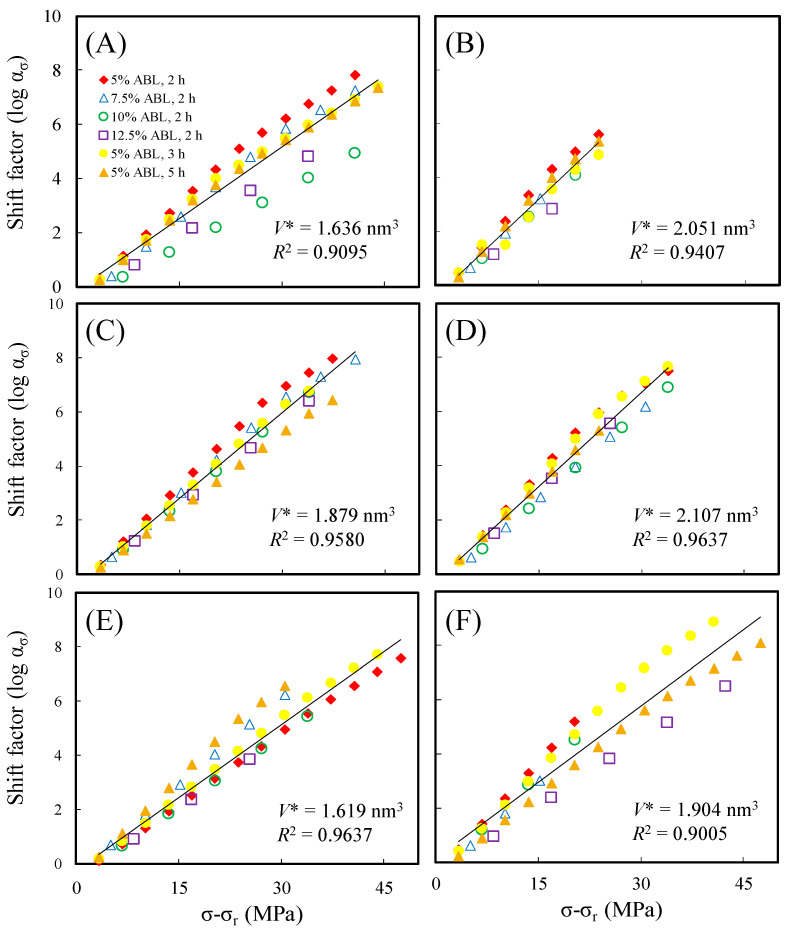
Typical Eyring equation plots of untreated (**A**), 160 °C/4 h (**B**), 160 °C/8 h (**C**), 160 °C/16 h (**D**), 180 °C/4 h (**E**), and 180 °C/8 h (**F**) hydrothermally treated Japanese cedar wood.

**Table 1 polymers-15-04149-t001:** Equilibrium moisture content (EMC), moisture excluding efficiency (MEE), modulus of elasticity (MOE), and modulus of rupture (MOR) of untreated and hydrothermally treated Japanese cedar wood.

Sample Code	EMC (%)	MEE (%)	MOE (GPa)	MOR (MPa)
Untreated	13.9 ± 0.3 ^A^	–	6.8 ± 0.8 ^A^	59.7 ± 9.6 ^A^
160 °C/4 h	7.8 ± 0.5 ^B^	44.4 ± 3.4 ^C^	6.4 ± 1.1 ^A^	53.1 ± 6.6 ^A^
160 °C/8 h	7.4 ± 0.5 ^B^	47.1 ± 3.5 ^C^	5.8 ± 0.8 ^A^	55.7 ± 8.2 ^A^
160 °C/16 h	7.3 ± 0.3 ^B^	47.9 ± 2.2 ^C^	6.8 ± 1.0 ^A^	58.7 ± 11.6 ^A^
180 °C/4 h	7.8 ± 0.3 ^B^	43.6 ± 1.8 ^C^	6.8 ± 0.6 ^A^	54.3 ± 7.0 ^A^
180 °C/8 h	7.2 ± 0.6 ^B^	48.6 ± 4.0 ^B^	6.2 ± 0.9 ^A^	57.4 ± 7.8 ^A^
180 °C/16 h	6.4 ± 0.3 ^B^	54.6 ± 2.2 ^B^	6.2 ± 1.1 ^A^	50.5 ± 9.8 ^B^
200 °C/4 h	7.9 ± 0.3 ^B^	44.0 ± 2.3 ^C^	6.0 ± 0.8 ^A^	49.7 ± 8.8 ^B^
200 °C/8 h	6.9 ± 1.2 ^B^	50.7 ± 8.7 ^B^	6.5 ± 0.7 ^A^	43.3 ± 8.4 ^B^
200 °C/16 h	5.4 ± 0.4 ^C^	61.5 ± 2.5 ^A^	4.9 ± 0.7 ^B^	29.1 ± 6.4 ^C^

Values are the mean ± SD (*n* = 13). Different letters within a column indicate significant differences at *p* < 0.05.

**Table 2 polymers-15-04149-t002:** Predicated creep compliances of untreated and hydrothermally treated Japanese cedar wood.

Specimen	*S*_0_ (GPa^−1^)	*a*	*b*	*S*(*t*) (GPa^−1^)					ICR (%)				
				Time (Years)					Time (Years)				
				1	5	10	20	50	1	5	10	20	50
Untreated	0.144	0.0011	0.27	0.16	0.16	0.17	0.17	0.18	–	–	–	–	–
160 °C/4 h	0.195	0.0020	0.31	0.23	0.25	0.27	0.28	0.31	−46	−53	−57	−62	−70
160 °C/8 h	0.177	0.0006	0.28	0.18	0.19	0.19	0.19	0.20	−18	−15	−14	−12	−10
160 °C/16 h	0.185	0.0007	0.26	0.19	0.20	0.20	0.20	0.21	−23	−20	−18	−16	−13
180 °C/4 h	0.139	0.0015	0.15	0.15	0.16	0.16	0.17	0.17	3	3	4	4	5
180 °C/8 h	0.162	0.0014	0.14	0.17	0.17	0.17	0.18	0.18	−9	−6	−4	−2	1

*S*(*t*) = *S*_0_ + *ae^bt^*, where *S*(*t*) is the time-dependent compliance value, *S*_0_ is the instantaneous elastic compliance value, *t* is the elapsed time, and *a* and *b* are constant values.

## Data Availability

The data presented in this study are available on request from the corresponding author.

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
