# Peer review of "Assessing the Long-Term Creep Behaviour of Hydrothermally Treated Japanese Cedar Wood Using the Short-Term Accelerated Stepped Isostress Method"

_polymers, 2023, doi:10.3390/polym15204149_

Round 1

Reviewer 1 Report

This work reports Assessing the Long-Term Creep Behaviour of Hydrothermally Treated Japanese Cedar Wood Using the Short-Term Accelerated Stepped Isostress Method. The manuscript structure is well organized. However, Some comments are suggested to the authors to improve the manuscript. For example:-

1.      In line 85, the author should mention the samples' dimensions and the distilled water's volume.

2.      There is no explanation of the advantages of the recent work on the published work in the introduction. And also, there is no information about the application of this work.

3.      There is no information about EMC, MEE, MOE, and MOR analysis in the text. However, the author has summarized the values of them in Table. The author should add the curve for flexural properties. 

Minor editing of the English language is required.

Reviewer 2 Report

Well done paper. Please look to the attachment concerning a point in the equations.

Reviewer 3 Report

The article is suitable for the journal but there are some aspects to improve:

-EMC (%), MEE (%), MOE (GPa), MOR (MPa), define and explain and clarify very carefully these features in your article

-Which are the main goals of the article? Define that

-Which the main conclusions of your research? Can be applied this materials in some industrials or other applications?

- Figure 1 is a very bad quality, please improve the quality of this figure

- Which is the novelty or novelties of this article? And which are the main contributions? (Please ass this information in the main text)

Round 2

Reviewer 1 Report

Accept

Author Response

Thank you very much for your kind assistance.

Reviewer 3 Report

The article has been ammended satisfactorily, but there is one aspect that I ca not see the improvements in the revised version, this is:

Figure 1 is a very bad quality, please improve the quality of this figure. 

The authors must provide a new version with clear improvements of this figure, because is difficult to see the diferences between different lines in  figure 1

Author Response

Figure 1 is composed of original curves under various SSM testing conditions, which is why the lines appear more complex compared to their fitted master curves in Figure 3. However, according to your suggestion, we have made every effort to improve Figure 1 quality in this revised version (Page 7).